# Overcoming Catastrophic Forgetting via Hessian-free curvature estimates

## Abstract

Learning neural networks with gradient descent over a long sequence of tasks is problematic as their fine-tuning to new tasks overwrites the network weights that are important for previous tasks. This leads to a poor performance on old tasks – a phenomenon framed as *catastrophic forgetting*. While early approaches use task rehearsal and growing networks that both limit the scalability of the task sequence orthogonal approaches build on regularization. Based on the Fisher information matrix (FIM) changes to parameters that are relevant to old tasks are penalized, which forces the task to be mapped into the available remaining capacity of the network. This requires to calculate the Hessian around a mode, which makes learning tractable. In this paper, we introduce Hessian-free curvature estimates as an alternative method to actually calculating the Hessian. In contrast to previous work, we exploit the fact that most regions in the loss surface are flat and hence only calculate a Hessian-vector-product around the surface that is relevant for the current task. Our experiments show that on a variety of well-known task sequences we either significantly outperform or are *en par* with previous work.

## 1 Introduction

The main goal of machine learning is the ability to generalize from the given training data to unseen examples. However, in practice the achievable degree of generalization is limited. While in the ideal case an end-to-end system learns complex functions from minimum input, it is often necessary to introduce a certain amount of prior knowledge. Such prior knowledge operates as an inductive bias and therefore has a constraining effect on the hypothesis space, i.e., the set of all possible functions that can be learned by the learning algorithm (Mitchell, 1980). While this sounds counter-intuitive such a reduction of the hypothesis space may lead to better generalization properties in practice (Mitchell, 1980). Hence, instead of eliminating the bias to increase generalization (as suggested by Hessel et al. (2019)), a promising direction of research tries to identify and introduce the right form of it.

We can achieve this by limiting the functions that can be expressed by the learning algorithm or by introducing bias to the learning algorithm itself. Simple examples include the choice for linear activations to only allow approximations of linear functions or to add a regularization term to the objective function. Similar to this, we can also improve generalization by training on different tasks (Baxter, 2000) from a task family at the same time or by introducing auxiliary tasks (Jaderberg et al., 2017). This is commonly known as multitask learning and has shown to not only improve generalization properties but also to be more sample-efficient (Baxter, 2000). Due to the limited availability of data for training we need a well-tuned inductive bias. Hence, such choices are crucial for the final real-world performance of any machine learning algorithm.

While *multitask learning* is a great tool to improve generalization and to reduce the amount of samples that are necessary to learn a family of tasks it is still limited in its scalability. Both the amount of tasks that can be learned and the amount of data required to learn them are strongly limiting factors. Consider, for instance, a reinforcement learning setup where an agent learns different tasks from interacting with in an environment. In practice we are limited in storing the data for all relevant tasks required to train a model on all tasks jointly. However, learning those tasks sequentially is also not an option as gradient descent and its variants (which are the dominant learning approaches for neural networks) do not consider the importance of individual parameters for early tasks.

This destructive learning is commonly termed as *catastrophic forgetting* (McCloskey & Cohen, 1989). While in the context of fine-tuning and pre-training (Erhan et al., 2009) this does not bear a problem (as the goal is not to reuse the previous parameter state, but rather to optimize the learning process for some target task) it becomes important in multitask problems where we wish to maximize generalization and sample-efficiency. It is also critical in the *continual learning* framework, where the parameters of a neural network are optimized over multiple datasets (representing different tasks) provided sequentially, which are not available at later time. The goal is hence to retain all (or most) of the important parameters for previous tasks and to be able to build-up on this knowledge for an arbitrary number of future tasks. Thus, the scalability of learning would only be limited by the capacity of the neural network but not by the properties of the training method.

The Bayesian framework (Kirkpatrick et al., 2017; Ritter et al., 2018) is a promising approach to address catastrophic forgetting. The information about former tasks is condensed in a prior, which not only preserves the knowledge about tasks but also introduces an inductive bias based on the learned tasks. *Elastic Weight Consolidation* (EWC) (Kirkpatrick et al., 2017) is a simple yet efficient way to reduce catastrophic forgetting. EWC approximates the prior with a Gaussian centered around the optimized network parameters for previous tasks, where the diagonal precision is given by the diagonal approximation of the *Fisher Information Matrix* (FIM). This approach has two significant downsides: i) each new task adds a new regularization term that penalizes changes of parameters that are relevant to previous tasks; and ii) the diagonal approximation of the FIM assumes independent network parameters, which leads to information loss with a growing number of tasks. Ritter et al. (2018) extend EWC but still approximate the prior from previous tasks using a Gaussian. They devise a block-diagonal approximation for the prior from the older tasks by defining a quadratic approximation whose solution requires to calculate the Hessian. The Hessian is in turn approximated by the block-diagonal *Kronecker-factored approximation*.

In this work we propose an alternative way of calculating the Hessian, based on well established *Hessian-free* (Schraudolph, 2002; Pearlmutter, 1994) methods to estimate curvature information of the network parameters. In contrast to Ritter et al. (2018), we exploit the fact that most regions in the loss surface are flat (Ghorbani et al., 2019). This allows us to use only a small subset of the Hessian as it holds enough relevant information. We then use a Hessian-vector-product to sample from this subset. This way, we can incorporate the importance of individual weights and include dependencies between the network parameters when we train the network over a long sequence of tasks. We evaluate our algorithm on permuted MNIST (Kirkpatrick et al., 2017), disjoint MNIST (Ritter et al., 2018) and single-headed disjoint MNIST (Farquhar & Gal, 2019), and compare with state of the art approaches. Our results show that we consistently outperform EWC across all tasks and that we are *en par* with Ritter et al. (2018) on the disjoint tasks, while our method has significantly lower space complexity compared to both EWC and Kronecker-factored approximation.

The remainder of this paper is structured as follows. Section 2 provides background on continual learning, EWC, and Kronecker-factored Laplace approximation. Section 3 describes our method in detail. Section 4 shows the efficiency of our approach and compares it against state of the art on a variety of well-known task sequences. Section 5 discusses related work. Section 6 concludes.

## 2 BACKGROUND

### 2.1 CONTINUAL LEARNING AND CATASTROPHIC FORGETTING

In the continual learning framework the parameters $\theta \in \mathbb{R}^n$ of a neural network are optimized over multiple datasets $\mathcal{D}_1, \ldots, \mathcal{D}_t, \ldots, \mathcal{D}_T$. These individual datasets become available to the training algorithm one after another and usually cannot be revisited at a later time. The goal is to achieve a high accuracy/performance on the current task (represented by the current dataset $\mathcal{D}_t$) while still preserving (high or most of) the performance for all the previously visited tasks. However, this is usually challenging for neural network models as commonly used gradient-based optimization methods cannot distinguish between important and unimportant parameters for previous tasks. As a consequence parameters that are relevant for previous tasks are modified (heavily), which leads to performance degradation when the network is used on any of those previous tasks (Rusu et al., 2016).

Hence, to address catastrophic forgetting in neural networks we need to retain the parameters that are important for previous tasks while still allowing the network to learn new tasks. However, at the same time we also want the space complexity of the network to be independent of the amount of tasks that were observed so far (and that are about to come). This means that learning a new task while retaining high performance on all prior tasks should be possible without adding new parameters or regularization terms for each new task, at least as long sufficient capacity is available. As a plus we want to foster some degree of parameter sharing to enable positive transfer effects, e.g., improved sample-efficiency due to the fact that past experience can be reused.

## 2.2 ELASTIC WEIGHT CONSOLIDATION (EWC)

EWC (Kirkpatrick et al., 2017) is a simple yet efficient approach that meets most of the above mentioned requirements. The key idea is to add a penalty when parameters that are important for previous tasks are about to be changed while parameters that are less relevant for previous tasks do not receive a penalty. EWC uses a quadratic penalty term that is derived from a Bayesian formulation of the problem (where all the information of all previous tasks is condensed in the prior) as follows:

$$p(\theta|\mathcal{D}_{1:t+1}) = \frac{p(\mathcal{D}_{t+1}|\theta)p(\theta|\mathcal{D}_{1:t})}{p(\mathcal{D}_{t+1})}, \tag{1}$$

where $p(\theta|\mathcal{D}_{1:t+1})$ and $p(\theta|\mathcal{D}_{1:t})$ are the posterior and prior distributions over the parameters $\theta$ of the network and $\mathcal{D}_1, \ldots, \mathcal{D}_t, \mathcal{D}_{t+1}$ are the datasets corresponding to the respective tasks. If we want to learn a new task we update the posterior by conditioning it on the newly available data $\mathcal{D}_{t+1}$.

However, we have to address two problems that stem from Equation 1. First, maintaining the full posterior over all previous datasets is usually intractable (Ritter et al., 2018; Opper & Winther, 1998) and we instead need to approximate it. Second, without storing the information from all previous tasks there is no easy solution to update the posterior.

The first problem can be addressed by approximating the posterior with a Gaussian (MacKay, 1992):

$$p(\theta|\mathcal{D}_{1:t}) \sim \mathcal{N}(\mu_t, \Sigma_t). \tag{2}$$

With two tasks A and B and their datasets $\mathcal{D}_A$ and $\mathcal{D}_B$, for the posterior $p(\theta|\mathcal{D}_A)$ the mean $\mu_A$ is given by the solution for the previous task $\theta_A^*$, and the precision $\Sigma_A^{-1}$, i.e., the inverse of the covariance, by the diagonal of the *Fisher information matrix* (FIM) $F$. Learning tasks A and B consecutively then results in the following objective function:

$$\mathcal{L}(\theta) = \mathcal{L}_B(\theta) + \frac{\lambda}{2}(\theta - \theta_A^*)^T F(\theta - \theta_A^*), \tag{3}$$

where $\mathcal{L}_B(\theta)$ is the loss depending on the current data $\mathcal{D}_B$, and $\lambda$ is a hyperparameter that controls the influence of the regularization term. At this point we only need to store the previous weights and the diagonal approximation of the FIM for the previous task. For another task C we store a separate FIM for that new task together with the solution for task B $\theta_B^*$, and add another regularization term:

$$\mathcal{L}(\theta) = \mathcal{L}_C(\theta) + \frac{\lambda}{2}(\theta - \theta_A^*)^T F_A(\theta - \theta_A^*) + \frac{\lambda}{2}(\theta - \theta_B^*)^T F_B(\theta - \theta_B^*). \tag{4}$$

## 2.3 KRONECKER-FACTORED LAPLACE APPROXIMATION

The diagonal approximation of the FIM assumes the parameters to be independent, which is rarely the case in practice. Ritter et al. (2018) address this shortcoming by adopting the Bayesian online learning approach (Opper & Winther, 1998). As the prior $p(\theta|\mathcal{D}_{1:t})$ preserves all the information about the previous tasks recursively using the previous posterior as the next prior makes it possible to find a MAP-estimate $\theta^* = \arg\max_\theta p(\theta|\mathcal{D}_1, \ldots, \mathcal{D}_{t+1})$ sequentially. Due to the fact that the posterior conditioned on all previous tasks is intractable, a parameterization of the posterior $p(\theta|\mathcal{D}_{t+1}, w(t))$ with parameters $w(t)$ is introduced. To update this parametric approximate posterior requires two steps:

1. **Update Step:** in an update step the old approximative posterior $p(\theta|w(t))$ is used to perform an update using the Bayesian rule (see Ritter et al. (2018) for a detailed analysis):

$$p(\theta|\mathcal{D}_{t+1}, w(t)) = \frac{p(\mathcal{D}_{t+1}|\theta)p(\theta|w(t))}{\int d\theta' p(\mathcal{D}_{t+1}|\theta')p(\theta'|w(t))} \tag{5}$$

2. **Projection Step:** In a projection step the new posterior $p(\theta|\mathcal{D}_{t+1}, w(t))$ is projected onto the same parametric family as $p(\theta|w(t))$ (as they are usually not from the same parametric family):

$$q(\theta|w(t+1)) \approx p(\theta|\mathcal{D}_{t+1}, w(t)). \tag{6}$$

Similar to EWC the update step can be approximated by a Gaussian approximate posterior:

$$\mathcal{L}(\theta) = \mathcal{L}_{t+1}(\theta) + \frac{1}{2}(\theta - \mu_t)^T \Sigma_t^{-1} (\theta - \mu_t). \tag{7}$$

As before, the mean $\mu_t$ is given by the solution for the previous task $\theta_t^*$. Accordingly, the parameters $w(t)$ are given by $w(t) = \{\mu_t, \Sigma_t^{-1}\}$. The core improvement that this framework offers is encapsulated in the projection step: instead of adding a new regularization term for each new task, $\Sigma_t^{-1}$ is instead projected to $\Sigma_{t+1}^{-1}$ which then maintains information about all tasks up to task $t + 1$. Ritter et al. (2018) realize this by computing the Hessian around the most recent solution $\theta_{t+1}^*$, and adding it to the Hessians from all previous solutions:

$$\Sigma_{t+1}^{-1} = H_{t+1}(\theta_{t+1}^*) + \Sigma_t^{-1}, \text{ where}$$
$$H_{t+1}(\theta_{t+1}^*) = -\frac{\partial^2 p(\mathcal{D}_{t+1}|\theta)}{\partial\theta\partial\theta}\bigg|_{\theta=\theta_{t+1}^*} \tag{8}$$

This way information about previous tasks can be preserved while still limiting the storage requirements to a constant number of parameters. However, in practice this approach needs to store a set of parameters per task.

## 3 HESSIAN-FREE CURVATURE ESTIMATION

Previous approaches identify the most important parameters for each previous task and then prevent the modification of those parameters during the training of a new task. EWC uses the diagonal of the FIM while Ritter et al. (2018) use a Hessian approximated using the block-diagonal Kronecker-factored approximation.

We address the same problem but approach it differently. We build upon the intuition of meta-learning in general and from the *model-agnostic meta learning* (MAML) algorithm (Finn et al., 2017) in particular. MAML identifies model parameters that (upon modification) lead to faster learning for all tasks in a given task distribution. By defining a meta-learning objective and using available data *for all tasks in the task distribution* it learns network weights that will lead to faster learning and generalization in new tasks, if being used as a starting point for the optimization.

In our case, apart from the fact that we assume no access to samples from previous tasks, we invert the intuition behind MAML: we identify model parameters that are sensitive to changes in each task but instead of tuning these parameters to be a good *starting point* for the fine-tuning of all tasks, we penalize large changes to them, as this will deteriorate the performance of previous tasks.

In order to identify the important network parameters, i.e., parameters that upon being changed lead to a big change in the loss, we also use the Hessian matrix, but in contrast to the Kronecker-factored Laplace approximation we exploit the fact that most regions of the loss surface are flat (Ghorbani et al., 2019). This allows us to use only a small subset of the Hessian as this subset already holds enough relevant information. We then use a *Hessian-vector-product* to sample from this subset.

In essence, we need to estimate directions with high curvature as at those points we find the *important* weights of the network. However, any computation involving the exact Hessian for larger networks is infeasible in practice. Hence, it is key to find a good approximation of the Hessian while still preserving enough curvature information to determine which parameters are crucial for the previous tasks. Fortunately, as most regions in the loss surface are flat it is sufficient to only extract information about the few regions that exhibit a high curvature. Thus, instead of computing the full Hessian we compute a Hessian-vector-product, which is similar to sampling the curvature in the direction of *a given vector*. There are two important questions to answer here: (i) how to efficiently calculate the Hessian-vector product, and (ii) how to chose a *suitable* vector/direction.

An efficient Hessian-vector-product calculation was initially presented in Pearlmutter (1994) and has subsequently been used for several Hessian-free (also called truncated-Newton) optimization

methods (Schraudolph, 2002; Martens, 2010). The key idea is that the Hessian is not calculated explicitly. Instead, for a given vector $v$ the Hessian-vector-product $Hv$ is directly computed using *finite differences* (Martens, 2010) at the cost of a forward- and a backward-pass through the network (e.g., using algorithms such as back-propagation). The Hessian-vector-product is then calculated by (see Pearlmutter (1994) for the implementation details):

$$Hv = \lim_{\epsilon \to 0} \frac{\nabla f(\theta + \epsilon v) - \nabla f(\theta)}{\epsilon} = \frac{\partial}{\partial \epsilon} \nabla f(\theta + \epsilon v) \Big|_{\epsilon=0} \tag{9}$$

Given that the Hessian-vector-product can be computed as described above, the second question is how to choose the vector $v$ that defines the direction in which we sample the curvature. Inspired by *Stochastic Meta-Descent* (Bray et al., 2004a;b), which uses the combination of the momentum and a Hessian-vector-product to estimate gradient directions with low curvature, our first choice to select the vector $v$ is to use the momentum. In our case the momentum is calculated using the exponentially weighted moving average of the past gradients:

$$v_{t+1} = \alpha v_t + (1 - \alpha) \nabla f(\theta), \tag{10}$$

where $\alpha$ controls the discount of older observations. The momentum is a sensible choice for the vector as it holds information about the parameters that have been changed the most during the training. The assumption is then that exactly these parameters will be among the most important ones for the most recent task. As such, if the parameters for the previous task $\theta_{t-1}^*$ are at an optimum, any change to important parameters results in a performance drop.

An alternative to the momentum is the eigenvector corresponding to the largest eigenvalue. This eigenvector represents the direction of highest curvature, and therefore by definition includes the most important parameters for the most recent task. A simple way to compute this eigenvector is to use the power method (Wilkinson, 1965), which entails computing a Hessian-vector-product.

Both versions result in a vector which maintains critical information about second-order interactions. From this vector we construct a positive semidefinite matrix by placing its absolute values as the entries of a diagonal matrix. Let $h_t$ be the resulting vector of the Hessian-vector-product $Hv$ for task $t$, then our curvature estimate $C_t$ is given as:

$$C_t = \begin{bmatrix} |h_{t,1}| & & \\ & \ddots & \\ & & |h_{t,n}| \end{bmatrix}, \tag{11}$$

with $n$ the number of network parameters.

The projection step then is defined as:

$$\Sigma_t^{-1} = C_t + \Sigma_{t-1}^{-1}, \tag{12}$$

and the final objective function for a new task $t + 1$ as:

$$\mathcal{L}(\theta) = \mathcal{L}_{t+1}(\theta) + \frac{\lambda}{2}(\theta - \theta_t^*)^T \Sigma_t^{-1}(\theta - \theta_t^*) \tag{13}$$

Similar to Kirkpatrick et al. (2018) and Ritter et al. (2018) we add a hyperparameter $\lambda$ to control the influence of the regularization term on the overall loss, i.e., that controls how to weigh the importance of the previous tasks over the most recent task.

One of the main advantages of our approach is the low storage requirements. Following the analysis in Ritter et al. (2018), Kronecker-factor approximation approach requires that all Hessians for previous tasks are kept in memory and the same holds for EWC, as the diagonal approximation of the FIM for all previous tasks are required to learn each new task. Instead, our approach only needs to store two vectors with the same size as the network parameters independently of the size of the task sequence.

## 4 EXPERIMENTS

In our experiments, we compare both of our Hessian-free curvature estimations (eigenvector and momentum) to closely related methods, i.e.. EWC (Kirkpatrick et al., 2017) and Kronecker-factored

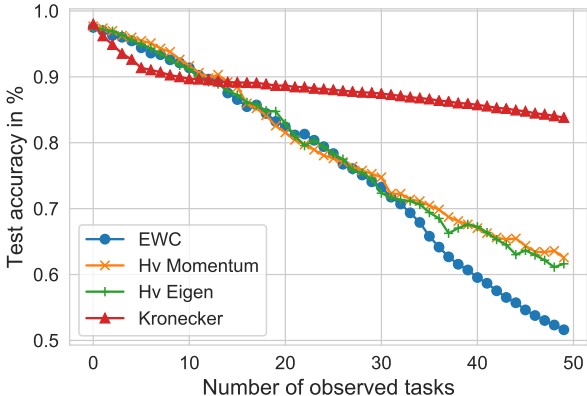

Figure 1: Experimental results using `permutedMNIST` for the best hyperparameters in comparison, and details of EWC, Kronecker factored approximation and Hessian-free with momentum and eigenvector.

approximation (Ritter et al., 2018). For both EWC and Kronecker-factored approximation we adapt the implementation from `https://github.com/hannakb/KFA`. We release the source code of our methods upon publication.

## 4.1 PERMUTED MNIST

For our first evaluation, we utilize the widely-used `permutedMNIST` dataset as presented in Goodfellow et al. (2013) and used in Kirkpatrick et al. (2017) and Ritter et al. (2018). The dataset contains $28 \times 28$ grey-scale images, that are permuted randomly in order to generate new tasks. Each permutation is a truly new task, since it is unrecognizable from its original.

For the evaluation, we perform a hyperparameter search with the following range of parameters: i) network structure: either 1 layer with 200 hidden units or 2 layers with 100 hidden units each; ii) $\lambda \in [1, 2, 3, 10, 20, 30, 100, 300]$. We use the ADAM optimizer with a learning rate of 0.001, a momentum of 0.5, and a batch size of 64 over 10 epochs.

Figure 1 shows the mean average accuracy over all 50 tasks with the best hyperparameters discovered for each method. While Kronecker-factor approximation achieves $83.82\%$, Hessian-free curvature estimation achieves $62.58\%$ and Hessian-free curvature estimation with the largest eigenvector achieves $61.63\%$, leading to better results compared to EWC ($51.62\%$) for the last 15 tasks.

Even though Kronecker-factored approximation achieves better performance compared to our approach, according to Farquhar & Gal (2019) in order to evaluate continual learning approaches other tasks can be more representative. In fact, Farquhar & Gal (2019) suggest to use a specific version of `disjointMNIST` which we evaluate below.

## 4.2 DISJOINTMNIST

For an evaluation according to the `DisjointMNIST` (Ritter et al., 2018) we split `MNIST` into two tasks: (1) letters '0' to '4' and (2) letters '5' to '9'. For this experiment we use a network with a ten-way classifier which makes the problem considerably more challenging than in the previous experiment where we used a five-way classifier. Hence, here the classifier learns a strong (bad) prior for the (respective) unseen classes in the datasets. It is more difficult as training on the second split can easily overwrite the parameters of the ten-way classifiers for the classes of the first split. We use a simple dense feed-forward network architecture with 2 layers and 100 hidden units in each layer as well as a batch size of 250 as reported in Ritter et al. (2018). We use 10 epochs and the same Adam parameters as in the `PermutedMNIST` experiment. This allows a comparison of our results against Kronecker-factored approximation and EWC.

Following the same evaluation procedure from Ritter et al. (2018) Figure 2a illustrates the result of a hyperparameter search over $\lambda \in [10^0, 10^1, \ldots, 10^7]$ for EWC, Kronecker-factored approximation,

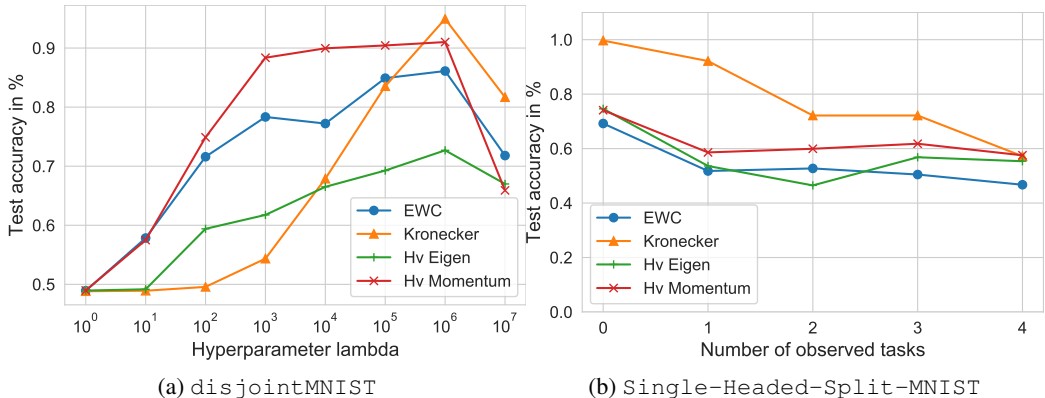

(a) `disjointMNIST`

(b) `Single-Headed-Split-MNIST`

Figure 2: Influence of $\lambda$ on different approaches.

and ours (i.e., Hessian-free curvature estimation using either the largest eigenvector or the momentum to estimate $v$). The results show the balancing of retaining information on old tasks over the learning accuracy on new tasks. Note that the different scales in $\lambda$ between our results and that from Ritter et al. (2018) only stem from different implementation details (but the results are still comparable). Similar to the `PermutedMNIST` experiment, we see that our approach (using the momentum) outperforms EWC with $91.01\%$ (at $\lambda = 10^6$) vs $86.11\%$ (which is what we expected as EWC disregards parameter dependencies that are not reflected by the diagonal of the FIM). Surprisingly, our approach is even comparable to the Kronecker-factored approximation (which reaches $94.93\%$) although our method uses considerably less storage memory to store information on the importance of parameters. The use of the largest eigenvector on the other hand performs poorly compared to the other methods with $72.69\%$ for $\lambda = 10^6$.

### 4.3 SINGLE-HEADED SPLIT MNIST

For the `Single-Headed-Split-MNIST` task (Farquhar & Gal, 2019) the available digits are split into five groups (i.e., tasks) of two classes each. The classifier (as for the `PermutedMNIST`) uses ten outputs, i.e., one for each digit, and the network is trained on each task one after another. In contrast to some other work (Zenke et al., 2017) all the tasks share the classifier head instead of having multiple task-specific outputs. Hence, the predictions are made for all possible outputs, not only for the outputs of classes that belong the most recent task.

We use the same network as in the previous experiments (i.e., 2 layers of 100 hidden units each) and a batch of 64. Figure 2b shows the results after a hyperparameter search over $\lambda$. As in the previous experiments we can observe that both of our Hessian-free curvature estimations consistently outperform EWC (Hessian-free with momentum achieves $57.54\%$ and the eigenvector approach $55.36\%$ while EWC reaches $46.73\%$) and that the momentum-based variant even comes again close to the Kronecker-factored approximation (which is at $57.2\%$ at the end).

## 5 RELATED WORK

Related work around the field of *catastrophic forgetting* is mainly driven by regularization methods, rehearsal methods, and dynamic architecture methods.

**Regularization Methods.** Elastic Weight Consolidation (Kirkpatrick et al., 2017) measures the distance between the network weights for the current task and the weight state of previous tasks, and applies a quadratic penalty weighted by a *diagonal* approximation of the Fisher information matrix to ensure that the new weights are not *too far* from the old weights. EWC only penalizes important parameters while the parameters that have no influence on the performance of previous tasks are allowed to change freely. Similar approaches have been proposed by Aljundi et al. (2018) and Lee et al. (2017). The main difference is how the importance of parameters for previous tasks are approximated. However, all these approaches have limited performance as they do not consider interactions between the parameters. Instead of using the diagonal of the Fisher information matrix (Ritter et al.,

2018) apply a Kronecker-factored approximation of the Hessian. This leads to strong improvements over EWC. This approach is most similar to ours, as it attempts to capture second-order parameter interactions to regularize parameter change. The main difference to our method is the usage of the Kronecker factorization to store the Hessian in a compact way while we exploit the fact that most regions in the loss surface are flat (Ghorbani et al., 2019). This allows us to use only a small subset of the Hessian as it holds enough relevant information. We then use a Hessian-vector-product to sample from this subset.

**Rehearsal Methods.** Rehearsal methods attempt to reduce catastrophic forgetting by replaying examples of previous tasks when learning a new task. A first approach here is to not only learn the actual task at hand but also the distribution of the training data. When a new task is learned, artificial samples from this learned distribution are added to the current set of training data. Typically this is done by adding a Variational Autoencoder (Kamra et al., 2017). Recent approaches (Shin et al., 2017) also employ generative adversarial networks with promising results. A second, more direct approach preserves a subset of the training data for each task in an episodic memory and reuses it to constrain the learning process of future tasks (Lopez-Paz et al., 2017). However, while being effective in reducing catastrophic forgetting in general, both approaches have shortcomings as the inherent problem of catastrophic forgetting is simply shifted to a scalability problem. In generative approaches samples for all previous tasks must be replayed each time to preserve old parameter states and as the number of tasks increases this becomes problematic. Similarly for the direct approach, even if only a small subset of examples for each task is preserved, still we can end up with a large dataset as the number of tasks increases.

**Dynamic Architecture Methods.** Another way to address catastrophic forgetting is to incrementally increase the capacity of the architecture. Approaches vary mainly in whether new capacity is added for each new task by default, or whether this is determined by a metric. Progressive Neural Networks (Rusu et al., 2016) add a new network for each new task and each new network is connected via lateral connections to the old ones to allow for transfer from previous tasks to the current one. This avoids catastrophic forgetting by design but as each new task requires a new network this approach does not scale well with the number of tasks. In contrast to Progressive Nets other approaches only add capacity when it is necessary. Part & Lemon (2016) present an approach based on Self-Organizing Map, which employs a similarity metric to determine whether a new node should be added to the network. Similar to this, Xiao et al. (2014) start out with a classifier with one super class and add new parameters, based on an error signal. Depending on the error made by the current model, only the final layer is extended by another output dimension, or a whole new sub-network is added as a subclass. Yoon et al. (2018) use the combination of sparsity and breadth-first-search to determine which parameters should be retrained for the current task. If the features learned so far are not able to represent the new task, more capacity is added dynamically (as in Xiao et al. (2014)). While these methods suffer significantly less from scalability issues, their main disadvantage lies in the fact that they have very stringent architectural constraints, which cannot be easily transferred to any arbitrary existing model.

## 6   CONCLUSION

This paper addressed catastrophic forgetting within a continual learning framework where the ultimate goal lies in the identification of the network weights that are important to previously learned tasks. While previous work in this direction is either limited in the achievable accuracy (as it only considers the diagonal of the Fisher Information Matrix) or limited in number of tasks (as they need to store information that grows linearly with the number of tasks) we set out to provide a first approach that uses second-order parameter dependencies with constant space complexity. We exploit the fact that most regions in the loss surface are flat, which allows us to use only a small subset of the Hessian as it holds enough relevant information. We then use a Hessian-vector-product to sample from this subset. This way, we can incorporate the importance of individual weights and include dependencies between the parameters when we train the network over a long task sequence.

We evaluated our algorithm on three widely used benchmarks and compared it with state of the art. Our results show that we consistently outperform EWC across all benchmarks and that we are better or at least *en par* with Kronecker-factor approximation, while our method at the same time requires significantly less memory.

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
