# OpenReview forum: "Overcoming Catastrophic Forgetting via Hessian-free Curvature Estimates"
_ICLR.cc/2020/Conference — Reject_

### Official Review · AnonReviewer1 · 2019-10-21
**Official Blind Review #1**

**Rating:** 1

**Review:**

1. Summary:
The paper considers neural network training in the continual learning setting -- data arrive sequentially and we can not revisit past data. The paper proposes an approximate Laplace’s method, in which the Hessian the log likelihood of the data is approximated by some form of Hessian-vector project (? - I will get to this question mark below). The paper considers some benchmark continual learning datasets and compares the proposed approach to EWC and Kronecker-factored online Laplace. The performance of the proposed approach is similar to that of EWC and worse than Kronecker-factored Laplace in most cases. Another sales pitch that the paper brings up a lot is the low space complexity, however this benefit has not been fully demonstrated, given the small-scale network/experiments.

2. Opinion and rationales

I’m leaning towards “strong reject” as I think the presentation needs another round of polishing and that the technical contributions need to be clarified / unpacked. I explain my thinking below.

a. The presentation/explanation/flow are not clear.
The abstract does not read well. For example: “This requires to calculate the Hessian around a mode, which makes learning tractable. In this paper, we introduce Hessian-free curvature estimates as an alternative method to actually calculating the Hessian.” This sentence makes it sound like current approaches are tractable, so what this paper is trying to address? The technical summary is also not precise, the Hessian-free methods used in the paper is to compute Hessian-vector products, not the actual Hessian.

The introduction motivates the continual learning problem using generalisation of neural networks leading to the need for multi-task learning; however multi-task learning is not scalable given the large number of tasks and thus we need to learn sequentially. However, I find this motivation not clear: if multi-task learning and its scalability issue are the reasons why we need continual learning, with the scale of the experiments considered in the paper, wouldn’t it always more beneficial to use multi-task learning instead of continual learning?

The prior work section is also not clear, in my opinion. The paper starts out by describing EWC as Bayesian updates and cites MacKay (1992), then talks about the Kronecker-factored Laplace approximation as “address this shortcoming by adopting the Bayesian online learning approach”, as if these methods are very different while in fact, these methods are some variants of the Laplace approximation, with different ways to approximate the Hessian. The issues described in section 2.2 “two problems that stem from eq 1” are not very clear, for example, “without storing the information from all previous tasks there is no easy solution to update the posterior” (?). I would follow the presentation/explanation in Ritter et al (2018), Huszar (2018) [a note on the quadratic penalty of EWC] and section 5 of the variational continual learning paper (Nguyen et al 2018) to provide a more succinct connection between these methods.
The connections between this work and MAML in section 3 is not clear to me. The continual learning and meta learning settings are also quite different.

b. The technical contribution is not clear and if correct, if of limited novelty.

What is not clear from reading section 3 is what quantity is being approximated, at what point a Hessian-vector product appears and thus we can use Hessian-free methods to approximate it. The paper talks about flat loss surface and sampling a small subset of the Hessian -- I’m not sure I understand these connections. In eq 11, the paper replaces the Hessian values with results of the Hessian-vector-product approximations -- this seems very odd to me, especially in terms of semantics and units, Hessian and hessian-vector-products are two very different things. Again, it is perhaps just me not understanding what is being approximated in the first place. The technical contribution of this paper is thus limited: using Hessian-free methods to approximate Hessian-vector products in the continual learning context.

c. The performance of the proposed method is not super exciting. Pragmatically speaking, it is not clear why practitioners should be using this in the near future given Kronecker-factored Laplace works and scales well in practice and there are a plethora of other recent methods (e.g. VCL) that are also developed from the Bayesian principle and work much better than EWC.


3. Minor details:

a. In eq 1, the denominator should be p(D_{t+1} | D_{1:t}).

b. Figs 1 and 2, I would use the same colour scheme throughout to be consistent.


**Experience Assessment:**

I have published one or two papers in this area.

**Review Assessment: Checking Correctness Of Derivations And Theory:**

I carefully checked the derivations and theory.

**Review Assessment: Checking Correctness Of Experiments:**

I carefully checked the experiments.

**Review Assessment: Thoroughness In Paper Reading:**

I read the paper thoroughly.

---

### Official Review · AnonReviewer2 · 2019-10-23
**Official Blind Review #2**

**Rating:** 3

**Review:**

The paper focuses on alleviating the problem of "catastrophic forgetting", exhibited by neural networks learned with gradient-based algorithms over long sequence of tasks. In such learning scenarios, tuning of parameters over the new tasks lead to degradation of performance over the old tasks as the parameters important for the latter are overwritten. The gradient-based algorithms are unable to distinguish between the important and  the not-so-important parameters of the old tasks. Hence, one direction of works, including the proposed one, aim at identifying the most important parameters for all the old tasks and discourage modifications on those parameters during the training of the new tasks.

Existing works like Elastic Weight Consolidation (EWC) (Kirkpatrick et al., 2017) have proposed a Bayesian framework to lessen such forgetfulness by condensing the information of the previous tasks and supplying it as a prior for the new task. In such a framework, Ritter et al. (2018) propose a quadratic approximation of the prior which requires computing (an approximate block-diagonal Kronecker-factored) Hessian.

The paper employs a recent result (Ghorbani et al., 2019) to argue that most regions of the loss surface are flat. Hence, computing the Hessian in only a few regions (which exhibit high curvature) should suffice. However, computing the exact Hessian for large networks is infeasible in practice. The paper, therefore, uses Hessian-vector-product (Schraudolph, 2002; Pearlmutter, 1994), which is similar to sampling the curvature in the direction of a given vector. The key advantage of the proposed approach is the low storage requirements. Regarding how to chose a suitable direction/vector, the paper suggests two choices: the momentum vector or the eigenvector corresponding to the largest eigenvalue (of the Hessian). The  motivation behind the above choices, especially the former option, is unsatisfactory. Empirically, we observe that the momentum vector is a better option than the eigenvector. However, a (theoretical/empirical) deep-dive into why momentum vector is a good candidate should be done.

Empirically, the proposed approach with momentum vector performs better than EWC but worse than Ritter et al. (2018). More discussion into the results (esp. Hv-momentum vs Hv-eigenvector) would have shed more light on the proposed approach.

**Experience Assessment:**

I do not know much about this area.

**Review Assessment: Checking Correctness Of Derivations And Theory:**

I assessed the sensibility of the derivations and theory.

**Review Assessment: Checking Correctness Of Experiments:**

I assessed the sensibility of the experiments.

**Review Assessment: Thoroughness In Paper Reading:**

I read the paper thoroughly.

---

### Official Review · AnonReviewer3 · 2019-10-24
**Official Blind Review #3**

**Rating:** 3

**Review:**

This paper proposes a method for tackling catastrophic forgetting. Similar to previous methods such as EWC (Kirkpatrick et al., 2017), they penalize parameter updates that align with the Fisher information matrix of the previous tasks. This will prevent the model from changing the previously useful parameters. They try to match the result of previous fisher-based methods but at a lower computational cost. They propose using a low-rank approximation to the Hessian using Hessian-vector-product with two types of vectors: the momentum velocity vector and the largest eigen-vector of the hessian. Then they build a diagonal approximation to the Hessian.

Cons:
- Eq 11, there is no justification for forming a curvature matrix by putting the absolute value of the hessian-vector-product with the proposed vectors on the diagonal. Particularly considering the largest eigen-value, Hv will be a vector of zeros with exactly one 1. This does not seem to be a good estimate of the hessian.
- Fig 1, the proposed method seem to perform poorly compared to the kfac-based method on permuted mnist.
- Figure 2 mainly compares to EWC as a baseline. In Farquhar & Gal (2019), other methods such as VGR perform significantly better. The proposed method is not competitive with state-of-the-art.

**Experience Assessment:**

I have read many papers in this area.

**Review Assessment: Checking Correctness Of Derivations And Theory:**

I assessed the sensibility of the derivations and theory.

**Review Assessment: Checking Correctness Of Experiments:**

I assessed the sensibility of the experiments.

**Review Assessment: Thoroughness In Paper Reading:**

I read the paper at least twice and used my best judgement in assessing the paper.

---

### Author Response · Authors · 2019-11-15
**Rebuttal Comment**

We would like to thank the reviewers for their feedback. Unfortunately, we are not able to address all comments to the extent and depth we would like to within the rebuttal period, but we will use the feedback as a guideline for improving on our paper and results and resubmit in the future.

---

### Decision · Program_Chairs · 2019-12-19

**Decision:**

Reject

**Comment:**

The reviewers have provided thorough reviews of your work. I encourage you to read them carefully should you decide to resubmit it to a later conference.